# Environmental and Social Influences on the Behaviour of Free-Living Mandarin Ducks in Richmond Park

**DOI:** 10.3390/ani12192554

**Published:** 2022-09-24

**Authors:** Camille Munday, Paul Rose

**Affiliations:** 1Royal Veterinary College, University of London, 4 Royal College Street, London NW1 0TU, UK; 2Centre for Research in Animal Behaviour, Psychology, University of Exeter, Perry Road, Exeter EX4 4QG, UK; 3Wildfowl & Wetlands Trust (WWT), Slimbridge Wetland Centre, Slimbridge GL2 7BT, UK

**Keywords:** *Aix galericulata*, wildfowl, time-budget, activity pattern, welfare

## Abstract

**Simple Summary:**

Collecting information on how wild animals behave in the free-living environment can be useful for improving how such species are managed when under human care (e.g., in the zoo). Mandarin Ducks are an example of a species with a large captive population where research into the behaviour of wild birds can help with explaining and evaluating how this species is coping in captivity. This research collected data on free-living Mandarins in a large public park and compared such data to published research on captive Mandarins to evaluate any differences in time budgets. The overall aim of this research was to provide information on what behaviours are commonest amongst free-living Mandarin Ducks to help others with the assessment of behavioural normality of captive birds.

**Abstract:**

Many species of birds are housed in zoos globally and are some of the most popular of animals kept under human care. Careful observations of how species live and behave in their natural habitats can provide us with important knowledge about their needs, adaptations, and internal states, allowing identification of those behaviours that are most important to the individual’s physical health and wellbeing. For this study, Mandarin Ducks (*Aix galericulata*) were chosen as a study species because, like many species of waterfowl, they are widely kept in both private institutions and zoos, yet little research has been conducted on their core needs in captivity. A free-living population of naturalised Mandarin Ducks living in Richmond Park was used for this research. Data on state behaviours (resting, swimming, foraging, perching, preening, and vigilance) were collected five days a week (08:00–18:00) from the 26 March to 26 May 2021. Secondly, temporal, seasonal, environmental, and animal-centric factors (e.g., Sex) were recorded to assess any impact on the Mandarin’s time-activity budget. Lastly, a comparison between free-living anmd captive activity was conducted (via the literature) to evaluate whether captive behaviours differ to how they are expressed in the wild. Results showed that free-living Mandarins predominantly rested (19.88% ± 28.97), swam (19.57% ± 19.43) and foraged (19.47% ± 25.82), with variations in activity related to factors such as vegetation cover and pond size. Results also showed differences between the time-budgets of free-living and captive Mandarins, suggesting that captive birds may not always have the opportunity to express species-typical behaviours. This research indicated that study of natural behaviours performed in the wild may help to evaluate “normal” behaviour patterns of zoo-housed individuals and provide evidence for environmental and husbandry alterations that can promote good welfare. However, any potential impact on the activity patterns of free-living species due to human interactions should be considered when assessing deviations between the behaviour of wild and captive individuals.

## 1. Introduction

Time-activity patterns of captive animals can be calculated and evaluated to understand how species’ respond to their captive environment and to infer animal welfare states [1,2]. Evidence on natural behaviours performed in the wild can help evaluate “normal” behaviour patterns of zoo-housed individuals and provide evidence for environmental and husbandry alterations. Modern zoos and aquariums (hereafter “zoos”) have come a long way from ancient menageries, where animals were kept in barren cages with very little stimulation. Instead, modern zoos strive to provide optimal husbandry conditions and the highest welfare standards for their animals [3,4]. However, without key biological evidence, captive animals may not thrive, and likely to experience a reduced overall quality of life [5]. This is because, with careful observations of how species live and behave in their natural habitats, we can gain important knowledge about their needs, adaptations, and internal states allowing us to identify those behaviours that are most important to the individual’s physical health and emotional well-being [5,6,7,8]. Maintaining those natural behaviours captive animals *want* or *need* to carry out, is not only vital to their wellbeing [9], but also to the success of conservation efforts and maintenance of sustainable populations [10,11]. This is because phenotypic and genetic divergence between wild and captive populations may occur if captive animals are required to adapt to their artificial environment and therefore any fitness benefits accrued from wild behaviour patterns may be lost [12].

As the majority of zoo-based behavioural and welfare research focuses on charismatic mammalian species [13,14,15,16], correct husbandry and management practices for many avian species are unknown. On top of this, knowledge of the welfare needs (and how to determine an individual’s welfare state) for some of the most widely kept non-mammalian species is lacking [13,15,17,18]. The Mandarin Duck (*Aix galericulata*) for example, is one species of duck that has become one of the most popularly kept of all wildfowl (Anseriformes), with a widespread aesthetic appeal due to its brightly coloured plumage and therefore it has a long history of captivity [19,20]. Despite such popularity, only a few behavioural studies have been conducted on this species [21,22] and consequently a baseline for “good Mandarin Duck welfare” is currently unavailable.

Native to both China and Japan [20], the Mandarin Duck (hereafter “Mandarin”) is considered a “perching duck” species [23]. Perching ducks (although not a taxonomic group) are more arboreal than other species of wildfowl and thus prefer to spend much of their time perched in trees [24]. Mandarins are most active in the morning and late afternoon but will also feed diurnally and nocturnally [21,22,24]. Although Mandarins primarily feed on aquatic plants and a variety of seeds, in the spring it also feeds on aquatic invertebrates [24] as ducks attaining breeding condition require more protein [20,25,26]. In the spring, female Mandarins have been noted as feeding for far longer periods of time than male birds [21]. While a Mandarin pair will breed for several seasons in a row, they do not necessarily mate for life [24]. Nevertheless, they form strong seasonal pair bonds where drakes are very protective of their partner [24], especially during the breeding season in spring [21].

Animals respond to environmental stimuli by altering their behaviour to improve their chances of survival and reproduction [27]. Temporal—i.e., time of day [21,22,28], seasonal [29,30]) and environmental factors (including pond size [31]) all affect how birds will partition their time to different behaviours. Social period (i.e., the different sections of the Mandarin’s breeding season when different social behaviours are performed) [21], prevailing weather conditions [28,32,33,34,35], vegetation structure [29,36], the presence of humans [37,38] and other species of bird [39] also influence behaviour patterns. Finally, individual animal factors such as energy requirements and sex-specific breeding activity [21,22] affect the time-budgets of birds and thus influence the breeding success and survival. Therefore, a better understanding of how such temporal, seasonal, environmental, and animal-centric factors influence behavioural performance and motivational states will aid in improved captive care [40,41,42,43].

The aims of this study were: (i) to quantitively study how free-living Mandarins allocate their time between different activities during spring; (ii) analyse any effect of sex, time of day, pond size, vegetation coverage, social period, weather, human presence, and number of other birds on this species’ time-activity budget; (iii) compare the time-activity budgets of free-living and captive Mandarins to determine whether activity levels and behavioural performance observed in captivity differs from that expressed in the wild. We predicted that sex differences in behaviour would be apparent due to differing investment in reproductive activity and that free-living Mandarins were likely to be more active than captive birds due to their wider choice of habitat usage and lack of behavioural restriction.

## 2. Materials and Methods

### 2.1. Sample Population and Study Site

Free-living (hereafter “wild”) Mandarins occurring in Richmond Park, London, UK were observed for this research. Richmond Park (51.4412° N, 0.2745° W) has a total area of 1012 hectares [44], covering woodland with ancient trees, acid grassland and various wetland habitats [45] that are all frequented by the Mandarins. For ease of sampling (due to the scale of the Park), Mandarins frequenting a specific woodland in the Park (the Isabella Plantation) were sampled; the Isabella Plantain is a 16.18 hectare enclosed woodland situated in the middle of Richmond Park [46]. Mandarins on and around two separate ponds of differing sizes (Peg’s Pond—Figure 1 and Thompson’s Pond—Figure 2) within the Isabella Plantation were the specific focus for data collection. From observations over several years, Royal Parks estimated that 35 breeding Mandarin Ducks regularly frequent this woodland (A. Ergun, personal communication, 26 March 2021) but the Mandarin population of the overall Park is likely much higher. Public feeding of the wildlife in Richmond Park (including the Mandarins) is actively discouraged by The Royal Parks [47] and the birds were not managed, being free to leave Richmond Park and frequent other habitat areas.

Prehistoric evidence suggests that the Mandarin was once a native bird in the UK but went extinct after the Pleistocene [48]. The first import of a Mandarin, in modern time, to the UK occurred around 1745 [48] and the species became established (naturalised) in the UK in 1930s. In 1971, in recognition of the stable population of Mandarins in Great Britain, the species was placed on the British List [48] (the definitive list of birds occurring in the wild in Britain as maintained by the British Ornithologist’s Union [49]). The current British population estimated at 5000 breeding pairs with a winter population of around 7000 birds [50]. Although the current population is (re)introduced, research has concluded that naturalised Mandarins in southern England have no invasive or detrimental impact on native species as they fill a previously unoccupied ecological niche [51], however if the species spreads north it may compete with other tree-nesting ducks, such as the Common Goldeneye, *Bucephala clangula* [52].

### 2.2. Data Collection

Behavioural observations of wild Mandarins were conducted during spring from 26 March 2021 to 26 May 2021, five days each week (from 08:00–18:00), giving a total of 38 days of observations. A three-day period of preliminary observation was instigated before the main data collection period commenced. Based on these observations and information from published literature [22,53,54] an ethogram of key behaviours was created (Table 1). Ponds were chosen based on their size (Peg’s Pond was larger and contained an island), habitat structure (degree of plant growth and cover, as shown in Figure 1 and Figure 2), and location with the Isabella Plantation (Peg’s Pond was near to the Plantation’s perimeter, Thompson’s Pond was located more centrally).

A single researcher collected all behavioural data. Methods were reviewed by the Ethics Committee of the Royal Veterinary College for MSc Wild Animal Biology students in the Spring Term 2021.

Instantaneous scan sampling [55] for one male and female Mandarin pair (defined as birds in close proximity, mirroring each other’s actions or conducting mutual preening indicative of a social bond) was used to record Mandarin state behaviours (Table 1). These behaviours were recorded at one-minute time intervals over a 20 min observation period. Multiple 20 min observations were conducted within three time periods (08:30–10:00, 13:00–15:00, 16:00–18:00). If both Mandarin sexes were not present at any one time or were not in a pair, one or two females or males were observed instead. In conjunction to this, continuous event sampling was used to record event behaviours (Table 1). To minimise selection bias, for every 20 min observation period, two new ducks were selected at random. This was achieved by waiting for five minutes, facing away from the pond, between each 20 min observation period. Moreover, to further minimise selection bias, and to reveal the effects of vegetation coverage on this species’ time-activity budget, a die was used to randomly select whether two ducks in open water (numbers 1 or 2), close to vegetation cover (numbers 3 or 4), or in a tree (numbers 5 or 6) were going to be selected. When needed, binoculars (8 × 42) were used to get a better view of the ducks. However, if at any point, a duck could no longer be tracked or seen, its behaviour was recorded as “Out of Sight”. As ducks were not individually identifiable, pseudoreplication may have occurred with the same bird being observed multiple times on the same day of observation even though the researcher randomised selection of each individual for each recording period as best possible.

To analyse the effect of time of day, observations were made in three time periods, (08:30–10:00, 13:00–15:00, 16:00–18:00). To analyse the effect of pond size, observations for each time-period alternated between Peg’s Pond and Thomson’s Pond. To analyse the effect of social period, a Mandarin’s breeding season (i.e., the entire observation period) was divided into three social periods: 26 March–28 April was classified as the Pre-Laying period, when male and female pairs were performing reproductive behaviours (see ethogram, Table 1); 28 April–7 May was classified as the Laying period, when very few female ducks were present due to nesting; 7–26 May was classified as the Post-Incubation period, when ducklings were present. Social Period categories were taken from Bruggers and Jackson [21] to allow for consistency with published literature. To reveal the effects of human and bird presence, visitor number, number of other Mandarins aside of the focal birds, and number of other waterfowl present were recorded at the start of every 20 min observation period. Other waterfowl included Canada Geese (*Branta canadensis*), Mallard (*Anas platyrhynchos),* Tufted Duck (*Aythya fuligula*) and Common Moorhen (*Gallinula chloropus*). Lastly, to reveal the effects of weather, weather description (Sunny, Sunny Intervals, Light Cloud, Thick Cloud, Light Rain and Heavy Rain), temperature (°C), wind speed (m/s), likelihood of precipitation (%), and humidity (%) were also recorded, via BBC Weather iPhone application, at the start of every 20 min observation period. Descriptions of weather were taken directly from the application.

### 2.3. Data Analysis

Averages ± standard deviation (x% ± SD) of behaviours were first calculated. If a focal animal could not be observed for ≥60% of the observation period for any 20 min time-period, it was excluded from all analyses. To construct the activity-time budget of the Mandarin during the breeding season, the average percentage of time spent in each of the state behaviours was calculated. To facilitate the comparison of wild and captive Mandarin behaviour, several behaviours were combined: From the wild ethogram, “Resting” and “Perching” were combined as “Loafing”, and “Human Foraging” and “Natural Foraging” as “Foraging”, whereas from the captive ethogram, “Swim Feed”, “Graze” and “Eat Corn” were all combined as “Foraging” [21]. Using data from Pre-Laying, Laying and Post-Incubation periods of the published graph (Figure 1, page 88), the average time (%) that captive Mandarins spent loafing, and foraging, between the hours of 08:30–18:00, was extrapolated from Bruggers and Jackson [21]. This time-period was chosen to match the data collection period for the direct observations of the wild Mandarins in Richmond Park.

Statistical analyses were performed in IBM SPSS v.26 (IBM, Armonk, NY, USA) for Windows [56]. Firstly, behavioural data on both foraging and loafing were normally distributed, and a one-sample t-test was thus used to determine differences between wild and captive time budgets. However, behavioural data categorised under Activity and Inactivity (Table 1) were not normally distributed, and thus a Wilcoxon signed-rank test was used instead, to compare the two related samples. The mean of each behaviour under comparison were calculated from the wild Mandarin data and compared to the published means for behaviour of the captive birds.

Generalised Estimating Equations with a negative binomial log link function were used to analyse the relationship between all the response variables (i.e., the average percentage ducks engaged in each state behaviour and the frequency of occurrence for each event behaviour) and fixed effects (i.e., sex, time of day, pond size, vegetation coverage, social period, number of visitors and other birds, and weather parameters). Date and the interaction with time was included to account for repeated sampling events. Individual identification of birds would have strengthened the application of GEE by removing pseudoreplication. These tests were chosen based on their versatility of use with small populations with potentially repeated methods [57]. Exchangeable working correlation matrix was used to account for the correlation between observation pairs. Each of the fixed effects were evaluated individually first and those with *p* value < 0.2 were further evaluated in multivariable models using a backward elimination approach. Lastly, a Post hoc Pairwise Comparison was performed to compare the different categories within a significant fixed effect. Multivariable analyses could not be performed on infrequently observed state behaviours (reproduction and flight), due to the insufficient amount of data and therefore results for these behaviours are only presented descriptively. Statistical results from Generalised Estimating Equations are also expressed as medians ± interquartile range (x% + IQR) as well as rate ratios (RR) with 95% confidence intervals, with *p* < 0.05 considered to be significant. 

## 3. Results

### 3.1. Time Budget

A total of 701 observations were analysed (females accounting for 230 of these, and males 471). The average number of Mandarins observed per session was 4 with a range of 0 to 13. Fewer observations were obtained for females because of their absence while incubating. Resting, foraging (natural foraging and feeding by visitor), swimming, perching, preening and vigilance, were the main state behaviours of the Mandarins over the spring and breeding season (Figure 3). Terrestrial locomotion and reproduction were the state behaviours which occurred the least (Figure 3). The median time Mandarins were observed flying was 0% ± 0. The maximum number of occurrences of flight as an event was 6 counts per 20minute observation period and as a minimum, 0 counts per 20 min period. Lastly, throughout the entire observation period Mandarins spent more time being active (25% ± 80) than inactive (5% ± 45; Z = −10.46, *p* < 0.001).

### 3.2. Sex Effects on Behaviour

As described above, females and males shared predominant state behaviours of resting, swimming, foraging, perching, preening and vigilance (Figure 4). In females, these behaviours accounted for 96% ± 141% of their total behaviours, and in males, 94% ± 136%. Females had a higher rate of natural foraging than males, and a lower rate of vigilance than males (Table 2). Lastly, vocalisation occurred less frequently in females than males (Table 2).

### 3.3. Habitat, Temporal, and Social Variables

There was no significant difference in activity levels between time of the day (Morning: 25% ± 80; Afternoon: 25% ± 85; Evening: 30% ± 80; *p* = 0.436). Throughout the entire observation period, a total of 379 and an average of 6 ducks were observed at Peg’s Pond whereas a total of 322 and an average of 5 Mandarins were observed at Thomson’s Pond. Ducks had a higher rate of swimming at Peg’s Pond than Thomson’s Pond (Table 2). Since only five ducks were observed in a tree throughout the entire observation period, these observations were removed from analyses. Ducks in open water had a higher rate of vigilance than those close to vegetation cover (Table 2). Conversely, ducks in open water had a lower rate of resting than those close to vegetation cover (Table 2). Lastly, ducks spent more time being active when in open water than when near vegetation cover (Table 2).

Although univariable analyses showed that Mandarins had higher rates of foraging during the Pre-Laying period (7.5 ± 30) than the Post-Incubation period (0% ± 20; RR = 1.26, 95% CI = 0.44–1.73, *p* < 0.001), multivariable analyses showed the opposite whereby Mandarins had lower rates of foraging during the Pre-Laying period than the Post-Incubation period (Table 2). Further analyses, performed to explore this change in direction, showed that *Sex* and *Social Period* were confounded and showed that although females had higher foraging rates during the Pre-Laying period than the Post-Incubation period, for males this was the opposite (Table 3). The same analyses also showed that females had higher foraging rates during the Laying period compared to both the Pre-Laying and Post-Incubation period (Table 3).

However, all analyses showed that ducks had a significantly higher rate of natural foraging during the Pre-Laying period than the Laying period (Table 2). Analyses also showed that ducks spent the most time being active during the Pre-Laying period than both the Laying and Post-Incubation period and the most time being inactive during the Laying period than both the Pre-Laying and Post-Incubation period (Table 2)

### 3.4. Weather Variables

Throughout the entire observation period Sunny Intervals and Light Cloud were the most common weather categories experienced. Ducks had a lower rate of natural foraging and swimming in the heavy rain than in all other weather types (Table 2). On the other hand, however, ducks a significantly higher rate of alertness in the light rain than in sunny intervals (Table 2). While Temperature only had a significant effect on swimming, likelihood of precipitation only had a significant effect on maintenance (Table 2). However, Wind Speed had a significant effect on both fleeing and flight (Table 2). Lastly, Humidity did not have a significant effect on any behaviour.

### 3.5. Effects of Humans and Number of Ducks on Behaviour

There was no impact of visitor presence on duck behaviour. Mandarin and other waterfowl number had a significant effect on natural foraging, vigilance and social interactions (Table 2). Mandarin number also had a significant effect on vigilance and interspecific interactions (Table 2).

### 3.6. Wild Ducks Compared to Captive Ducks

Although there does not appear to be a significant difference between the amount of time wild and captive Mandarins spend loafing during the Pre-Laying and Laying periods, wild Mandarins (30 ± 18.82) spend a significantly shorter amount of time loafing than their captive counterparts (85 ± 30) during the Post-Incubation period (t(11) = −9.41, *p* < 0.001) (Figure 5). On the other hand, however, wild Mandarins spend a significantly shorter amount of time foraging than their captive counterparts during both the Pre-Laying (Wild: 23.68 ± 6.96; Captive: 25.00 ± 41.13; t(11) = −5.15, *p* < 0.001) and Laying (Wild: 8.13 ± 9.13; Captive: 25.50 ± 33.75; t(11) = −13.31, *p* < 0.001) periods, but spend a significantly larger amount of time foraging during the Post-Incubation period (Wild: 15.31 ± 11.97; Captive: 0 ± 14.38; t(11) = 2.56, *p* = 0.027) (Figure 5).

## 4. Discussion

Overall, the daily time-activity budget of Mandarins in Richmond Park consisted of resting, foraging, swimming, perching, preening and vigilance, with variation in the performance of these behaviours related to sex, time of day, pond size, vegetation coverage, social period and number of birds present, and the weather. Resting, swimming, and foraging made up the highest proportion of the time-budget and this is consistent with other species of waterbird [58,59] and duck specifically [60,61,62,63].

### 4.1. Time-Activity Budget

Mandarins were significantly more active than inactive during the breeding season and they spent a higher proportion of their time active (63.37%) than those during the wintering period (58.21%) [22]. The slightly higher levels of foraging and activity observed during this season, may reflect seasonal changes in food availability (e.g., more animal-based food being selected), as reproduction in birds is timed to coincide with maximum food availability and therefore efficient foraging is important for reproductive success [25]. Female mandarins may be choosing higher energetic food items in preparation for eggs laying. Although perching did not make up the highest proportion of the time-budget, Mandarins still spent a relatively large amount of time performing this behaviour amongst tree branches. This is not surprising considering, as the name implies, they are “perching ducks” [24] and the need to perform this behaviour should be considered when housing this species in captivity.

### 4.2. Sex Effects on Behaviour

Mandarins showed a significant difference in time spent foraging and vigilance by sex. Captive Mandarins and other duck species have shown similar sexual differences during the breeding season [21,64,65,66]. Differential energy costs of reproduction are likely the cause of these variations in foraging rates because egg production is energetically costly [67] and thus breeding females require a high proportion of protein [20,25,26]. Sex differences in vigilance rates emphasise the importance of attendant males in protecting a female during foraging from other competition from other males [68]. This also explains why vocalisation occurred more frequently in males, as they attempt to ward off threats and/or signal about nearby danger [24].

### 4.3. Temporal and Environmental Factors

Time of day did not significantly affect levels of activity in these birds, contrary to published information [20,24]. However, unlike this study, these authors do not focus on one season alone. Therefore, these results could suggest that Mandarins remain active throughout spring days due to the demands of reproduction. Bruggers and Jackson [21] research supports this idea, as they showed that during Pre-Laying and Laying periods, Mandarin females were active throughout the day but extended their mid-day inattentive period during the summer and autumn.

Mandarin swimming rates were higher at Peg’s Pond than at Thomson’s Pond potentially due to the differences in pond size and habitat accessibility. Throughout the entire observation period, Peg’s Pond had a larger average (6 ± 2.84) and total number of observation (379) of Mandarins compared to Thompson’s Pond (5 ± 1.79 birds and 322 observations). Mandarins may have preferred to spend more time at Peg’s Pond due to its larger overall area and that it provided more resources to support different species of waterfowl. Peg’s Pond was surrounded by a littoral vegetation of common reed (*Phragmites australis*) and sedges (Cyperaceae), which is the Mandarin’s preferred shelter choice [24]. Mandarins appeared to frequently loaf on the island in the middle of Peg’s Pond. Mandarins can be wary of other species [24], and seek cover, thus ducks may have felt safer loafing on an island surrounded by water, vegetation, and trees. The results of this study showed that Mandarins closer to vegetation had higher resting and lower vigilance rates compared to those in open water. Therefore, providing captive Mandarins with adequate cover and dense vegetation in some enclosure zones is important for their comfort, reduced fear responses and for overall bird welfare. Although waterbirds can make defined habitat occupancy choices on a small scale and within short distances of different environmental resources [69], further study of Mandarin pond size preferences is suggested. Observations of ducks on ponds in different woodlands, with a larger distance between them would provide further information on Mandarin habitat choices and their preference for specific habitat types and structures.

Although female Mandarins had the highest foraging rates during the Laying period, due to the limited number of observed females during this period, no valid inferences could be drawn about differences in proportion of time spent foraging during this period compared to others. Nesting females only leave their nests to feed early in the morning or late in the evening [24], which would explain why so few were observed during the day throughout the Laying period. Future research should thus try to include observations of nesting females to increase the accuracy of the female Mandarin’s time-activity budget. The sexual differences in foraging rates between the Pre-Laying and Post-Incubation periods, as well as the higher rates of foraging during the Pre-Laying period than the Laying period, supports the observations of other authors- breeding females require high protein diets and thus consume a much greater proportion of aquatic invertebrates before egg laying [21,70]. The higher levels of activity during the Pre-Laying period than the Laying period reflects the importance of ducks having to remain active to maximise food intake in preparation for reproduction. On the contrary, levels of inactivity and resting rates were highest during the Laying period, most evidently because males were guarding their incubating females [24]. In fact, similar findings in resting rates have been reported for the same social period [21].

#### 4.3.1. Effects of Weather

Mandarins exhibited lower natural foraging and swimming rates in heavy rain compared to all other types of weather, most likely because they were seeking shelter. Periods of heavy rain were also associated with relatively strong winds whereby the average wind speed for heavy rain was 17.11 m/s. Since the onset of severe weather and strong winds can cause heightened stress in ducks [32], Mandarin behaviour likely changed as a result of seeking refuge from inclement weather. Alertness also occurred more frequently during rain compared to other types of weather, and there was a significant effect of wind speed on fleeing behaviour; for every 0.44704 m/s. increase in wind speed, there was a 1% increase in the occurrence of fleeing. As ducks are clearly responding to inclement weather conditions by seeking cover and increasing vigilance rates, captive husbandry must provide adequate shelter where Mandarins can retreat to so that they feel secure. Open enclosures without shelter or cover should be avoided and retreat areas built into such exhibits to provide cover from inclement weather.

Increased swimming rates in relation to falling temperature were consistent with other data in the literature [22,35,71]; for every 1 °C rise in temperature, swimming rates decreased by 3%. Although birds have evolved multiple mechanisms to regulate temperature in cold conditions, such as increasing their metabolic rate to maintain body temperature [72,73], maintaining higher body temperatures in colder weather is energetically challenging [74]. Harsh weather conditions can disturb a duck’s thermoregulation and metabolism, sometimes resulting in catabolism of lipid reserves and weight loss [74].

Precipitation had a significant effect on maintenance as maintenance rates increased by 1% for every 1% increase in precipitation. Although ducks have evolved to cope with heavy rain, i.e., sleeking their feathers to increase water proofing [75], a prolonged storm can cause hypothermia [76], which is why birds need to rest somewhere warm and dry during prolonged heavy rain. Basic duck welfare is improved when birds can thermoregulate correctly and preen, oil and clean their feathers [77]. Providing captive Mandarins with adequate shelters during heavy rain enables birds to cope, both behaviourally and physiologically, with periods of challenging weather.

#### 4.3.2. Effects of Human Presence and Number of Ducks

Visitor number did not have a significant impact on vigilance, alert or fleeing behaviour of these Mandarins, indicating that the Mandarins in the Isabella Plantation have become habituated to human presence. Specific individuals of specific species of birds in urban environments can display reduced flight initiation distances to human presence (and other potential threats too) [78]. Consequently, although wild, the Mandarins in Richmond Park may have habituated to human presence and thus reduced their vigilance rate and flight distance over generations of living within this human-created environment. Habituation to human presence has been recorded in other birds [30,79,80,81] and therefore suggests that the habitat enables the Mandarins to feel comfortable around humans (by providing adequate opportunities for escape) or because human behaviour reinforces the presences of the Mandarins. Habituation is most likely caused by visitors to the park regularly feeding the ducks. Frequent feeding and acclimating ducks to ”free” food, may lead to problems such as ducks relying on food from unnatural sources and poor bird health [82]. As a result, natural foraging behaviours as well as the loss of their innate fear of people may be curtailed with potential impacts on bird longevity and survival [82]. Thus, review of visitor effects on captive birds, especially those in walk through enclosures or with direct visitor contact, should be considered to prevent any abnormal changes in bird time activity patterns.

The number of Mandarins present had a significant effect on natural foraging and vigilance rates. For every increase in Mandarin number, natural foraging increased by 11%, vigilance by 10% and the frequency of conspecific social interactions by 29%. Social facilitation, where the performance of one individual’s behaviour increases the likelihood of others adopting the same behaviour or intensifying it [83], has been observed in ducks [84,85], including Mandarins when preening [21]. This may explain why foraging rates increased with the number of birds present. Highly sociable animals are more likely to exhibit social facilitated behaviour, which is thought to optimise resource exploitation and protection from predators [83]. Therefore, considering Mandarins are both highly sociable but shy birds [24], it seems logical that they prefer to forage in groups, maximising resource exploitation and feeling safer from predators. Number of waterfowl overall however (i.e., other species aside from the Mandarins), had an opposite effect on natural foraging; for every increase in the number of other waterfowl species, natural foraging decreased by 7%. This may be due to Mandarins attempting to avoid unnecessary competition with other species, especially during the breeding season. Housing Mandarins in a social group that facilitates social activities of benefit to each individual bird and considering mixed species exhibit to remove overt aggression or intense competition from other more outgoing species is required to maximise captive welfare.

### 4.4. Comparing Wild and Captive Mandarin Behaviour

The comparison between the time-activity budget of wild and captive Mandarins [21] suggested that activity patterns are more diverse and levels of activity higher in wild birds compared to published data on captive counterparts. The higher levels of foraging in captive Mandarins might be explained by the ease of access to food and lack of opportunities to perform other behaviours. Alternatively, the season that these wild Mandarins were observed (and the relative abundance of food present) may have allowed time for other non-foraging activities during daylight hours, with foraging potentially occurring outside of the observation times (i.e., during crepuscular or nocturnal periods). Higher levels of inactivity in captive Mandarins may be explained by this same scenario- birds do not need to actively seek out feeding patches. Higher levels of inactivity could also be a result of flight restraint (e.g., pinioning) that restricts the bird’s choice of locomotion behaviours and restricts behavioural performance to terrestrial and aquatic activities.

Moreover, flight restraint impacts a Mandarin’s ability to perch off the ground. Preventing captive Mandarins from being able to perch may result in chronic stress that could be detrimental to long-term bird health and wellbeing. It has been recorded that depriving captive birds of opportunities to perform highly motivated behaviours leads to stress-induced psychological problems, weight loss, changes to the immune system, and decreased reproductive capacity [86,87,88]. Therefore, captive management techniques that restricts a bird’s normal locomotory functions, such as flight, should be considered against the ecology of the species and potential long-term welfare implications [18].

Meta-analyses of wild behavioural data can provide a useful benchmark for assessing any behavioural differences under captive conditions [43] and such an approach supports the development of future research questions. Whilst comparison of these wild data with captive data is useful to the evaluation and analysis of behavioural indicators of welfare, differences in methods of data collection, influences of husbandry and environment of captive and the time of year for data collection need to be considered. Further study should repeat data collection with direct observation of captive birds alongside that on wild birds at the same time of year using the same observational methods. The limited data on flying (as both a state and event behaviour) recorded during this study shows the challenge in recording flying as a discrete and specific behavioural occurrence using traditional behavioural recording and sampling procedures. We encourage other researchers to extend this type of study, and perhaps use GPS [89] or accelerometery [90] technology or similar to fully determine how often, and for how long, ducks fly.

Finally, further study should consider ringing of individual free-living Mandarins to enable accurate identification of each bird within the population and hence removal of pseudoreplication and potential selection bias that may have occurred in our study, due to the lack of individual bird identification. The behaviour of some individual ducks may be overrepresented in the dataset if a specific behavioural characteristic, e.g., boldness or activity, drew the observer’s eye to them more often than to other birds in the same flock.

## 5. Conclusions

This study revealed the amount of time that free-living Mandarins spend resting, foraging and being vigilant to maximise survival and reproduction. Time-activity budgets were strongly influenced by numerous temporal, seasonal, environmental, and animal-centric factors. It also revealed that activity-patterns were more diverse, and levels of activity higher, in wild Mandarins compared to published data on captive birds. It is evident that both wild time-activity budgets and the factors that influence a Mandarin’s behaviour must be considered (i.e., moving toward more naturalistic food preparation and presentation methods, providing appropriate food for the breeding season, space to allow for flight, shelters for times of stormy and cold weather, vegetation cover for comfort and well-being, housing Mandarins in an appropriate social group, etc.) in order to manage captive Mandarins in a way that ensures good welfare. To understand more about behaviour measures of welfare, suitable for the evaluation of Mandarin behaviour in captivity, future research should include (i) observations of the behaviour of nesting female ducks, (ii) construction of a 24 h time-activity budget to illustrate behavioural change with circadian rhythm, and (iii) direct comparison of wild and captive Mandarin behaviour in different management and enclosure styles. These questions would help evidence the most appropriate care for this ever-popular species of waterbird.

## Figures and Tables

**Figure 1 animals-12-02554-f001:**
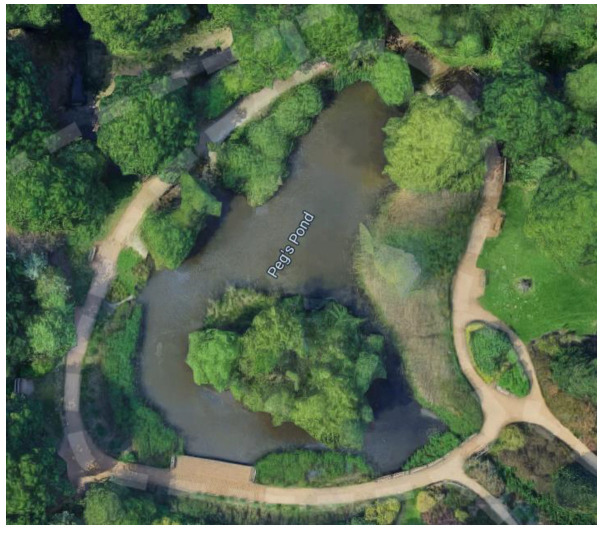
Google Maps view of Peg’s Pond with an area of open water of approximately 936 m^2^. Google (2022) Isabella Plantation, Richmond Park. Available online: http://maps.google.co.uk (accessed on 26 July 2022).

**Figure 2 animals-12-02554-f002:**
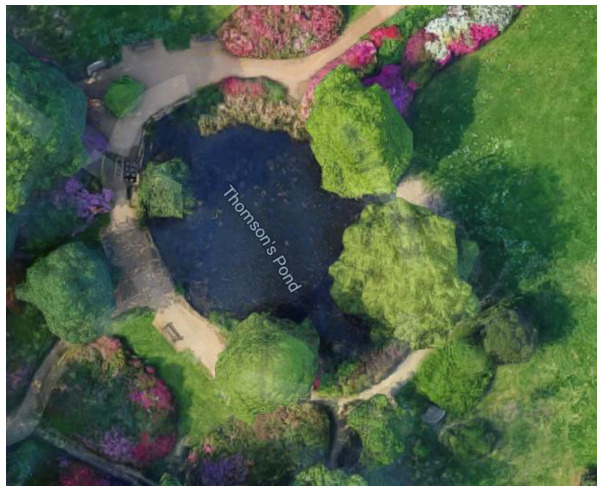
Google Maps view of Thomson’s Pond with an area of open water of approximately 439 m^2^. Google (2022) Isabella Plantation, Richmond Park. Available online: http://maps.google.co.uk (accessed on 26 July 2022).

**Figure 3 animals-12-02554-f003:**
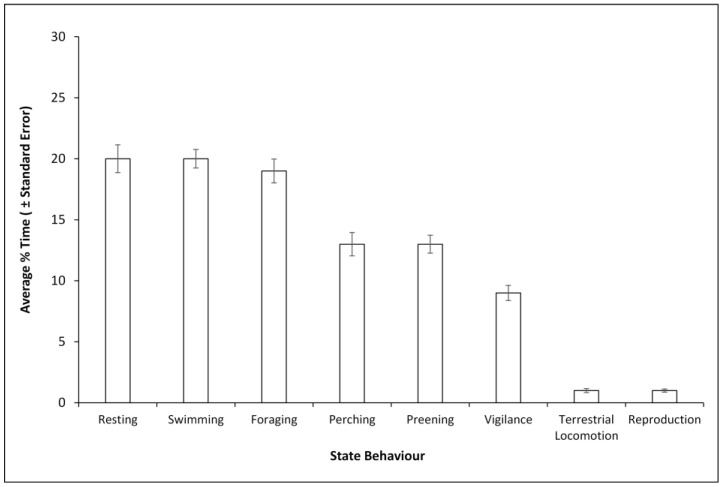
Average time budget (%) of Mandarins during the breeding season.

**Figure 4 animals-12-02554-f004:**
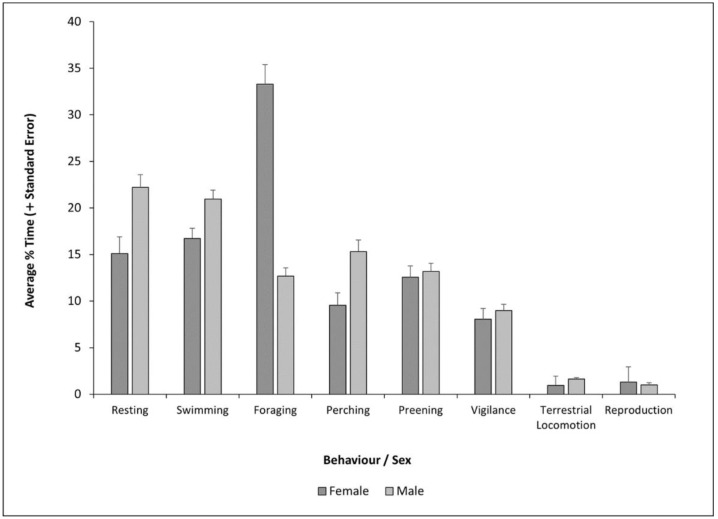
Average percentage time budget (showing positive SE bars) of the female and male Mandarin during the breeding season.

**Figure 5 animals-12-02554-f005:**
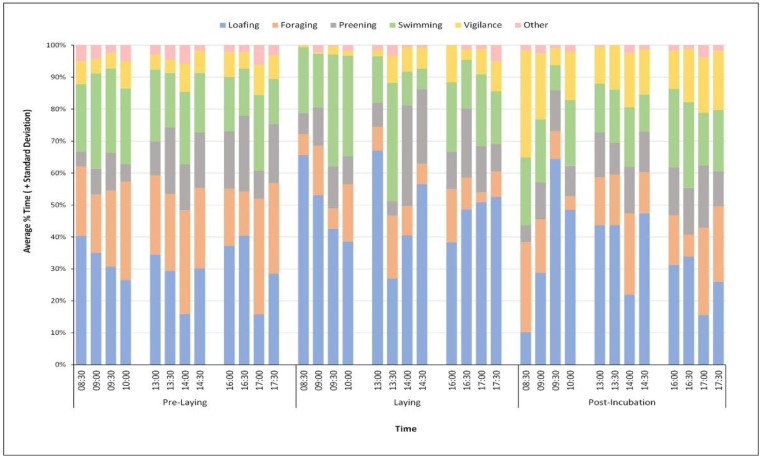
Average percentage time budget of the Mandarin during different social periods. Averages with standard deviations are tabulated in the Appendix A.

**Table 1 animals-12-02554-t001:** Definitions of various state and event behaviours of Mandarins. Behaviour patterns of short duration (<5 s long) were classified as “Events” and those of a long duration (>5 s long) as “States”. Behaviours were also grouped into active and inactive states whereby Active is characterised by the presence of motion. When a bird went out of sight this was also recorded.

Category	Behaviour	Description
	States	
Active	Natural Foraging	Surface feeding, diving feeding, catching, or swallowing food or duck seeks out food (naturally) in and out of water
	Feeding by Visitor	Surface feeding, diving feeding, catching, or swallowing food or duck seeks out food (from visitors) in and out of water
	Preening	Any element of the preening sequence including nibbling feathers, head rolls and shaking, that occurs when duck is either in the water or on land
	Terrestrial Locomotion	Slow or rapid movement on the ground, out of the water
	Swimming	Slow or rapid movement on the water with no foraging behaviour
	Reproduction	Courtship: Drake performs a drinking-preening-behind-the-wing sequence in the water/Hen nibbles the throat region of mate and utters coquette call. Hen flattens herself on the water in copulation posture, turning around and around Copulation: Drake is on top of a hen mating in the water
	Vigilance	Duck is stood upright, motionless, alert, and watchful, focusing on a particular alarming stimulus for a relatively long duration
	Flying	Flying for a relatively long duration, usually away from the study site
Inactive	Resting	Loaf or sleeping such as eyes are closed (or one eye is closed), neck is short, no head movements and/or or bill is tucked under wing on either water or land
	Perching	Loaf or sleeping such as eyes are closed (or one eye is closed), neck is short, no head movements and/or bill is tucked under wing in either a tree, on a branch or above ground
	Events	
	Vocalisation	Display call is like a thin, whistling and rapidly rising “hueessst, accompanied by deeper clappering sounds. Other sounds include a short and sharp, coot-like “ket”, and a short “ack”.
	Alert	Duck raises head and is attentive for a very brief moment during the performance of a state behaviour and then resumes the original behaviour after a few seconds.
	Fleeing	Duck is moving quickly away from another animal, usually in response to a threat or other aggressive behaviour
	Conspecific Social Interaction	Any brief interaction with another Mandarin including pecking, aggression or chasing
	Interspecific Social Interaction	Any brief interaction with another species of bird, including pecking, aggression or chasing
	Flight	Any brief flight which usually occurs within the study site
	Maintenance	Mostly body fluffing, body shaking and wing flapping but also sometimes stretching, scratching, flapping, bathing, head dip in water

**Table 2 animals-12-02554-t002:** Output of Generalised Estimating Equations including the Rate Ratios (RR) with 95% Confidence Intervals for significant factors and behaviours.

						95% Walk Confidence Interval for Difference	
Factor		Behaviour	Median ± IQR	*p* Value	RR	Lower	Upper
Sex	Female	Natural Foraging	20% ± 50	<0.001	3.53	2.88	4.34
	Male		0% ± 15		1		
	Female	Vigilance	0% ± 5	<0.001	0.08	0.46	0.13
	Male		0% ± 15		1		
	Female	Vocalisation	0% ± 0	<0.001	0.64	0.51	0.81
	Male		0% ± 0.00084		1		
Pond ID	Peg’s Pond	Swimming	15% ± 35	0.006	1.39	1.01	1.75
	Thomson’s Pond		15% ± 20		1		
Social Period	Pre-Laying	Natural Foraging	7.5 ± 30	0.006	0.54	0.35	0.84
	Post-Incubation		0% ± 20		1		
	Pre-Laying	Natural Foraging	7.5 ± 30	0.003	2.09	1.29	3.38
	Laying		0% ± 10		1		
	Pre-Laying	Activity	27.5% ± 80	<0.001	1.40	1.17	1.67
	Laying		20% ± 65.5		1		
	Pre-Laying	Activity	27.5% ± 80	0.020	1.30	1.05	1.60
	Post-Incubation		25% ± 95		1		
	Laying	Inactivity	25% ± 65	<0.001	1.51	1.21	1.89
	Pre-Laying		5% ± 35		1		
	Laying	Inactivity	25% ± 65	<0.001	1.65	1.26	2.16
	Post-Incubation		0% ± 36		1		
Vegetation Coverage	Open Water	Vigilance	0% ± 15	<0.001	1.82	1.41	2.34
	Close to Veg		0% ± 5		1		
	Open Water	Resting	0% ± 20	<0.001	0.48	0.37	0.62
	Close to Veg		15% ± 60		1		
	Open Water	Activity	35% ± 95	<0.001	1.96	1.66	2.30
	Close to Veg		10% ± 47.5		1		
Weather Description	Light Cloud	Natural Foraging	5% ± 25	0.016	0.10	0.02	0.65
	Light Rain		0% ± 15	0.023	10.10	.0.02	0.72
	Sunny		5% ± 29	0.025	0.11	0.02	0.76
	Sunny Intervals		5% ± 30	0.008	10.08	.0.02	0.51
	Thick Cloud		5% ± 36	0.032	0.09	0.01	0.81
	Heavy Rain		0% ± 0		1		
	Light Cloud	Swimming	15% ± 30	<0.001	0.21	0.09	0.50
	Light Rain		15% ± 40	<0.001	0.16	0.07	0.39
	Sunny		15% ± 20	<0.001	0.18	0.08	0.44
	Sunny Intervals		15% ± 20	<0.001	10.21	0.09	0.50
	Thick Cloud		15% ± 26	<0.001	0.25	0.09	0.72
	Heavy Rain		0% ± 7.5		1		
	Light Rain	Alertness	0 ± 0	0.036	1.69	1.04	2.76
	Sunny Intervals		0 ± 0.000833		1		
Temperature		Swimming		0.048	0.97	0.93	0.05
Precipitation		Maintenance		0.001	1.01	1.00	1.01
Wind Speed		Fleeing		0.015	1.01	1.01	1.10
		Flight		<0.001	0.90	0.84	0.97
Mandarin Duck N°		Natural Foraging		0.016	1.11	1.02	1.21
		Vigilance		0.028	1.098	1.010	1.193
		Interspecific Interaction		<0.001	1.292	1.170	1.427
Other Waterfowl N°		Natural Foraging		<0.001	0.93	0.90	0.97

**Table 3 animals-12-02554-t003:** Output of Generalised Estimating Equations including the Rate Ratios (RR) with 95% Confidence Intervals for female and male Mandarins for “Natural Foraging” during different Social Periods.

						95% Confidence Interval for Difference	
Behaviour	Sex	Social Period	Median ± IQR	*p* Value	RR	Lower	Upper
Natural Foraging	Female	Pre-Laying	30% ± 55	<0.001	2.54	1.65	3.91
		Post-Incubation	0% ± 15		1		
	Female	Laying	50% ± 60	0.034	1.76	1.21	2.56
		Pre-Laying	30% ± 55		1		
	Female	Laying	50% ± 60	<0.001	4.49	2.62	7.68
		Post-Incubation	0% ± 15		1		
	Male	Pre-Laying	5% ± 15	0.019	0.66	.0.45	0.95
		Post-Incubation	0% ± 23		1		

## Data Availability

Data is available upon reasonable request from the corresponding author.

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
