# Peer review of "Environmental and Social Influences on the Behaviour of Free-Living Mandarin Ducks in Richmond Park"

_animals, 2022, doi:10.3390/ani12192554_

Round 1

Reviewer 1 Report

The manuscript of Munday & Rose is an interesting piece of work. It will definitely add new notions to the literature by providing information on what behaviours are commonest amongst wild mandarin ducks to support the welfare assessment of their counterpart kept in captivity. The manuscript is well-written and of high quality, the discussion is clear and provides insights on some practical implications. Therefore, I highly recommend this paper for publication and I have only minor corrections to suggest as listed below:

1.      L 85: There is a missing bracket after the reference [31];

2.      L111: ‘the British and Irish List’. What type of list are the authors referring to? Please, clarify this aspect for those readers that are not familiar with it – the reference provided may not be a proper fitting (i.e. Shurtleff and Savage, 1996. The wood duck and the mandarin: The northern wood ducks), perhaps consider to look for a more suitable one;

3.      L 115: ‘as they fill an previously unoccupied..’ – Typo. Remove the ‘n’ from ‘an’.

4.   L131-133: Was a single researcher collecting the data? If so, please specify this aspect;

5.     L 171-183: It is not well clear how the authors identified/recognised each of the 35 mandarin ducks included in the study in order to avoid observing the same pair of ducks after the 5 minutes waiting and facing away from the pond.

6.    Figures 3 and 4: Are you presenting descriptive data or statistical results? In L 234-235 the authors state that ‘Statistical results are also expressed as medians ± interquartile range (x%+IQR) as well as rate ratios (RR) with 95% confidence intervals..’ however, results in your figures are presented as average % and SE. On the other side, if this is descriptive data, it should be presented as average % and SD instead of SE.

Author Response

Replies to reviewer 1

The manuscript of Munday & Rose is an interesting piece of work. It will definitely add new notions to the literature by providing information on what behaviours are commonest amongst wild mandarin ducks to support the welfare assessment of their counterpart kept in captivity. The manuscript is well-written and of high quality, the discussion is clear and provides insights on some practical implications. Therefore, I highly recommend this paper for publication and I have only minor corrections to suggest as listed below:

  1. L 85: There is a missing bracket after the reference [31];

Edited.

  1. L111: ‘the British and Irish List’. What type of list are the authors referring to? Please, clarify this aspect for those readers that are not familiar with it – the reference provided may not be a proper fitting (i.e. Shurtleff and Savage, 1996. The wood duck and the mandarin: The northern wood ducks), perhaps consider to look for a more suitable one;

Thank you for the comment. The British & Irish List is managed by the British Trust for Ornithology. It is the definitive record of birds that live in the British Isles and is quoted in the Wood Duck and Mandarin text. We have added a description of what the British List is to clarify.

  1. L 115: ‘as they fill an previously unoccupied..’ – Typo. Remove the ‘n’ from ‘an’.

Edited

  1. L131-133: Was a single researcher collecting the data? If so, please specify this aspect;

Extra information included.

  1. L 171-183: It is not well clear how the authors identified/recognised each of the 35 mandarin ducks included in the study in order to avoid observing the same pair of ducks after the 5 minutes waiting and facing away from the pond.

Thank you for the comment. The overall population was estimated at 35 ducks by Royal Parks (the organisation that runs Richmond Park). We did not mark them. We have explained that sampling of the same bird may have occurred but we had no way of individually marking ducks. We have noted the potential for pseudoreplication and that others should consider marking the birds.

  1. Figures 3 and 4: Are you presenting descriptive data or statistical results? In L 234-235 the authors state that ‘Statistical results are also expressed as medians ± interquartile range (x%+IQR) as well as rate ratios (RR) with 95% confidence intervals..’ however, results in your figures are presented as average % and SE. On the other side, if this is descriptive data, it should be presented as average % and SD instead of SE.

Thank for you for the comment. These are the outputs of the Generalised Estimating Equations and we have stated so in the table titles for Table 2 and Table 3. We have also added this detail into the data analysis section.

Reviewer 2 Report

1. I read throughout the MS. Authors followed 35 individuals of mandarin duck in London, in two lakes. However, ducks were observed - although intensively - for only 2 months, thus producing non-representative results. This is a serious flaw which forces me to reject your work.

2. The species is native to CHina and Japan, but authors should also describe the introduced range of the species.

Author Response

I read throughout the MS. Authors followed 35 individuals of mandarin duck in London, in two lakes. However, ducks were observed - although intensively - for only 2 months, thus producing non-representative results. This is a serious flaw which forces me to reject your work.

Thank you for the comments. We are sorry that you do not feel the paper is relevant for publication. This is a case study on a free-living population that is not manipulated. We feel it provides useful information on wild duck behaviour that can be compared to captive birds for extending husbandry and welfare evidence.

The species is native to CHina and Japan, but authors should also describe the introduced range of the species.

We have explained the range of the species in the manuscript. 

Reviewer 3 Report

Ms-ID: animals-1860073

Review

Environmental and social influences on the behaviour of wild mandarin ducks in Richmond Park

Camille Munday, Paul Rose

The present study describes the activity budget and behavior pattern of a free-living population of mandarin ducks and determines the influencing environmental, individual and social factors, as well as differences with a captive-living population. The manuscript presents interesting information, and is well written and structured. Topics and objectives are well introduced and contextualized, and the interpretation of results is detailed, comprehensive and well documented. However, there are several major concerns that I believe should be addressed or explained before considering the manuscript for publication. Most of them are related to methodological issues, such as sample size, study design (e.g. predictors, data autocorrelation) and statistical modeling.

Abstract & Introduction

Authors made a good work presenting the context of the study, objectives and its justification. The introduction is well written and structured. The only issue to point out is that the aim of the manuscript is to describe the activity budget of wild-living mandarins and compare them with captive-living conspecifics, but the studied birds belong to an urban introduced population habituated to human presence and “regularly” fed. Thus, I am not sure if their behavior could be considered as the expected under actual wild conditions. Of course, I do not mean that “true wild” (no human influence) is optimal and thus preferable (in fact there is no such natural vs human dichotomy, but a gradient of human intervention). On the contrary, certain humanized habitats may provide extra resources and increase biological fitness of flexible species. That could be the case of free-living mandarins that voluntary select the study sites. In any case, authors could refine the context of the wild-living population used as reference in the study. 

Minor corrections:

L11 - Replace “to improving” with “to improve”.

L13 - “explaining and evaluating” instead of “explain and evaluate”.

L31-32 – Provide standard deviation for mean percentages.

Methods

This section is well structured and written, with abundant information on data collection and handling, as well as statistical methods used. However, there are some major concerns that should be clarified or modified, basically related with predictor variables and modeling.

Please, add a map showing the geographical location of Richmond Park within London area and, perhaps, an inset map with the general location at country scale (though we all know where London is).

According to Results, the mean number of simultaneous mandarins was only 5-6 birds in each pond (what was the range?). I suspect that those were usually the same birds or part of a small local population. Authors mention that the studied population consists of 35 breeding pairs, but they do not specify if this is only for the two ponds observed, the Richmond park (there are more and larger ponds), or they are part of a much larger population in London area (large ponds in western and northern outskirts). The probability of repeated measured on the same few birds is high, which is impossible to know unless birds were individually marked. Therefore, inference on a small net sample size could bias observed patterns due to individual (with-in-subject) non-independent data. This is related to further comments on GEE analysis, subject ID and correlation structure considered (see concerns on modeling).

The predictor “social period” tries to account for three different breeding stages throughout the two-month sampling period. However, I find the temporal categorization too arbitrary and far from the actual phenology and temporal variability in the species. For example, the mid stage of “laying”, that includes egg lying and incubation, was set from 28 April to 7 May. This is only 9 fixed days when the average incubation period for the species is 28-30 days, without considering the clutch completion period (1st egg to incubation) and inter-nest variability. Maybe the “laying” period corresponds to the observed peak of incubation activity, but then further details are needed. An alternative approach would be to consider a temporal structure (e. quadratic) as a covariate instead of discrete arbitrary stages to model non-linear behavior patterns. 

More explicit details on the captive-living dataset used are needed (e.g. source, sampling period and dates, sites, sample sizes, data recovery, etc.). They may be relocated to “Data collection” instead of “Data analysis”. As I understand, data on loafing and foraging activities in captive birds were hourly means between 8:30 and 18:00 throughout the whole three-year sampling period. And those, were compared to the raw data collected in the present study? Or with equivalent hourly means?

Authors applied Generalized Estimating Equations (GEE) instead of Generalized Linear Mixed Models (GLMM) which is fine when analyzing non-standard longitudinal data sets as in this case. However, there are some concerns on modeling procedures:

-      Pond size and vegetation coverage are included as predictors in the model. However, repeated observations where made only in two ponds, which means that there are no true replicates if considered as factors (e.g. large/small ponds) or enough sample size if considered as covariates. In addition, those ponds are not really independent because they are only 250 m apart. As best, you will have a trivial comparison between two specific non-independent ponds (Pegs’s vs Thomson’s) but not between actual size categories (or gradient). Therefore, I suggest to remove both predictors from the analysis.

-      Multicollinearity among predictors should be checked before running models. There are many predictors than could present inter-correlation and thus biasing estimate determination (e.g. change in direction for the effect of social period on foraging, L290-295). Correlation matrices and/or VIFs could be applied to check for collinearity and select predictors.

-      What is the grouping factor for repeated observations (I think named as subject ID in SPSS) in the model? Observed birds are not ringed (at least not mentioned) and thus cannot be individually identified between sampling periods. Is it pond identity?

-      An exchangeable correlation working structure is used in the GEEs, which implies a fixed correlation value among observations. Why don’t you apply a structure accounting for temporal autocorrelation in your longitudinal data? For example, an autoregressive with lag one (AR1) that assumes correlation between consecutive observations. In any case, a brief justification of the correlation structure chosen should be incorporated.

-      Backward selection of predictors based on p-values is not the best approach for model evaluation. Instead, I recommend multi-model dredging based on information-theory criteria adapted for GEE, such as Quasi AIC, that is, QIC values. I think packages as MuMIn in R can handle this type of (automatic) procedures when working with many predictors. In that case you don’t need to make univariate models to preselect predictors.

Other minor corrections:

L203 – Remove one “be”.

L216-218 – Was homogeneity of variance of groups tested before the t-tests (e.g. Levene’s test)?

L196-199 – I guess that weather information from BBC application provides forecast information and interpolated estimations instead of actual local values observed. For example, authors consider likelihood of precipitation (%) instead of actual precipitation recorded in the study area or nearby. Is this kind of information reliable? Why not a small portable weather station to record instant in-situ parameters? Or at least observed information from nearby weather stations?

Results

Expand the information in tables and figure captions to better understand the source of data. For example, I suppose results presented in Table 2 and Figures 4, 5 are from the GEEs (and not from univariate models).

I suggest to summarize the outcome tables from GEE models in supplementary material if not presented in the main text (maybe too much numbers), with coefficient estimates, errors, statistics, p-values, etc.

L286-295 – This result shows an interaction between social period and sex predictors on natural foraging. Authors could consider first order interactions in the models, though that could increase complexity and convergence problems in some cases.

L306-308 – Information related to predictor description (e.g. weather categories) must be included in Methods’ section and not in Results. Also, provide more details on how each category is defined (e.g. cloud coverage thresholds).

Discussion

The discussion is complete, neat, and well supported by previous knowledge and referenced literature. Concerns could arise from discussion of problematic predictors, such as pond size.

L346-349 – Higher foraging activity could be related also to higher energy demand during the breeding seasonal, not only to higher food availability. In fact, evolutionary pressures would lead to synchronized highly demanding breeding activities and resource pulses in seasonal environments. This could explain also why females showed higher natural foraging rates during laying and pre-laying periods, much stronger than in males.

L374-386 – Although most of the interpretations could be reasonable according to the species ecology, inference on pond size effects based on only to small non-independent sites with different habitat structure (e.g. vegetation coverage, islands) is not possible as already commented in Methods. At most, those can be collateral non-tested conjectures.

Author Response

Replies to reviewer

The present study describes the activity budget and behavior pattern of a free-living population of mandarin ducks and determines the influencing environmental, individual and social factors, as well as differences with a captive-living population. The manuscript presents interesting information, and is well written and structured. Topics and objectives are well introduced and contextualized, and the interpretation of results is detailed, comprehensive and well documented. However, there are several major concerns that I believe should be addressed or explained before considering the manuscript for publication. Most of them are related to methodological issues, such as sample size, study design (e.g. predictors, data autocorrelation) and statistical modeling.

Thank you for the comments and useful feedback. We have attempted to answer all of your queries below.

Abstract & Introduction

Authors made a good work presenting the context of the study, objectives and its justification. The introduction is well written and structured. The only issue to point out is that the aim of the manuscript is to describe the activity budget of wild-living mandarins and compare them with captive-living conspecifics, but the studied birds belong to an urban introduced population habituated to human presence and “regularly” fed. Thus, I am not sure if their behavior could be considered as the expected under actual wild conditions. Of course, I do not mean that “true wild” (no human influence) is optimal and thus preferable (in fact there is no such natural vs human dichotomy, but a gradient of human intervention). On the contrary, certain humanized habitats may provide extra resources and increase biological fitness of flexible species. That could be the case of free-living mandarins that voluntary select the study sites. In any case, authors could refine the context of the wild-living population used as reference in the study.

Thank you for the comments. We have edited the end of the abstract based on your suggestions.

Please note that we have explained that these mandarins are free-living and that they are in an urban setting. We have included information in the discussion that urban birds may have different responses to humans and we have included description in the methods of the population and how those running these parks (“Royal Parks”) attempt to keep wildlife wild. These ducks are still wild and can find their own food and move out of the park at their own free will.

 Minor corrections:

L11 - Replace “to improving” with “to improve”.

Edited

L13 - “explaining and evaluating” instead of “explain and evaluate”.

Edited

 L31-32 – Provide standard deviation for mean percentages.

Thank you for the comment. We have removed the numeric results from the abstract to just state the three most common behaviours.

Methods

This section is well structured and written, with abundant information on data collection and handling, as well as statistical methods used. However, there are some major concerns that should be clarified or modified, basically related with predictor variables and modeling.

Please, add a map showing the geographical location of Richmond Park within London area and, perhaps, an inset map with the general location at country scale (though we all know where London is).

Thank you for the comment. We have included coordinates for the Park. We have included information from Royal Parks on where Richmond Park for others to go and look up (multiple references). We do not wish to add any further figures to the manuscript.

According to Results, the mean number of simultaneous mandarins was only 5-6 birds in each pond (what was the range?). I suspect that those were usually the same birds or part of a small local population. Authors mention that the studied population consists of 35 breeding pairs, but they do not specify if this is only for the two ponds observed, the Richmond park (there are more and larger ponds), or they are part of a much larger population in London area (large ponds in western and northern outskirts). The probability of repeated measured on the same few birds is high, which is impossible to know unless birds were individually marked. Therefore, inference on a small net sample size could bias observed patterns due to individual (with-in-subject) non-independent data. This is related to further comments on GEE analysis, subject ID and correlation structure considered (see concerns on modeling).

We have added in the average number of birds seen per day and the range too.

Thank you for the comments. The mandarins in Richmond Park fly in and out of the Isabella Plantation and can use all of the park. Royal Parks estimated (verbally to the observer) that, maximally, 35 birds have been seen in the Isabella Plantation when breeding. We have edited the methods to show that the population across the Park overall is likely to be much higher.  

We have already explained in the discussion the case study approach of this project but we still feel it is useful data on what these animals have done. We have no way of knowing if we have repeated measures or not but we have already acknowledged the need for marking and banding in the future (as a research extension). We have acknowledged this limitation (lack of individual bird ID).

The predictor “social period” tries to account for three different breeding stages throughout the two-month sampling period. However, I find the temporal categorization too arbitrary and far from the actual phenology and temporal variability in the species. For example, the mid stage of “laying”, that includes egg lying and incubation, was set from 28 April to 7 May. This is only 9 fixed days when the average incubation period for the species is 28-30 days, without considering the clutch completion period (1st egg to incubation) and inter-nest variability. Maybe the “laying” period corresponds to the observed peak of incubation activity, but then further details are needed. An alternative approach would be to consider a temporal structure (e. quadratic) as a covariate instead of discrete arbitrary stages to model non-linear behavior patterns.

Thank you for the comment. We took these categories from the literature and we have stated this in the methods for clarity and openness. We decided on these categories to allow for comparison with the published literature between our data and published data.

More explicit details on the captive-living dataset used are needed (e.g. source, sampling period and dates, sites, sample sizes, data recovery, etc.). They may be relocated to “Data collection” instead of “Data analysis”. As I understand, data on loafing and foraging activities in captive birds were hourly means between 8:30 and 18:00 throughout the whole three-year sampling period. And those, were compared to the raw data collected in the present study? Or with equivalent hourly means?

Thank you for the comment. We have clarified how data were compared between the published work and our observational data. We calculated the mean of the observed data and compared to the means of the published data.

Authors applied Generalized Estimating Equations (GEE) instead of Generalized Linear Mixed Models (GLMM) which is fine when analyzing non-standard longitudinal data sets as in this case. However, there are some concerns on modeling procedures:

-      Pond size and vegetation coverage are included as predictors in the model. However, repeated observations where made only in two ponds, which means that there are no true replicates if considered as factors (e.g. large/small ponds) or enough sample size if considered as covariates. In addition, those ponds are not really independent because they are only 250 m apart. As best, you will have a trivial comparison between two specific non-independent ponds (Pegs’s vs Thomson’s) but not between actual size categories (or gradient). Therefore, I suggest to remove both predictors from the analysis.

Thank you for the comments. The two ponds were different in scale of vegetation and size. One pond was located near at entrance to the Isabella Plantation (this is explained in the methods) and the other was more central. We feel that is it important to keep these two ponds in because they represent different choices for the birds. The advice sort on statistical analysis recommended keeping the two ponds separate.

-      Multicollinearity among predictors should be checked before running models. There are many predictors than could present inter-correlation and thus biasing estimate determination (e.g. change in direction for the effect of social period on foraging, L290-295). Correlation matrices and/or VIFs could be applied to check for collinearity and select predictors.

Thank you for the comment. We have followed the advice provided at the time on model selection and fit. We have discussed the potential of other aspects of weather, season and bird number already in the discussion. We have noted the small scale of the project and the potential limitations of this.

-      What is the grouping factor for repeated observations (I think named as subject ID in SPSS) in the model? Observed birds are not ringed (at least not mentioned) and thus cannot be individually identified between sampling periods. Is it pond identity?

Thank you for the comment. We have clearly stated that birds were not ringed and that we would consider this an important research extension in the future. We are not sure about your question about repeated measures because we attempted to record a different bird for each observation. Whilst not perfect, we cannot say that we were repeating observations as the number of mandarins would change with birds flying in and out.

We have included information on potential pseudoreplication already at the request of previous reviewers. This is found in the methods and in the discussion.

We have also included a further reference on the suitability of GEEs for this type of research project but if this is not suitable, we are happy to get in touch with our stats advisor who can provide more information. However, we will need an extension to the review. 

-      An exchangeable correlation working structure is used in the GEEs, which implies a fixed correlation value among observations. Why don’t you apply a structure accounting for temporal autocorrelation in your longitudinal data? For example, an autoregressive with lag one (AR1) that assumes correlation between consecutive observations. In any case, a brief justification of the correlation structure chosen should be incorporated.

We have responded to this comment below.

-      Backward selection of predictors based on p-values is not the best approach for model evaluation. Instead, I recommend multi-model dredging based on information-theory criteria adapted for GEE, such as Quasi AIC, that is, QIC values. I think packages as MuMIn in R can handle this type of (automatic) procedures when working with many predictors. In that case you don’t need to make univariate models to preselect predictors.

Thank you for the two comments above. We were provided with statistical advice on this from academic colleagues and we are happy with our approach to see any potential fit of the model. The information presented above way not suggested as a way of analysing these data. We have discussed all of the associated limitations with this dataset and we have encouraged others to extend the research with other methods if they feel it could be conducted in a more robust manner.

Other minor corrections:

L203 – Remove one “be”.

Edited

L216-218 – Was homogeneity of variance of groups tested before the t-tests (e.g. Levene’s test)?

We were not instructed to do this further based after testing for the distribution (normality) of these data that were used in either the t-test or WIlcoxen test.

L196-199 – I guess that weather information from BBC application provides forecast information and interpolated estimations instead of actual local values observed. For example, authors consider likelihood of precipitation (%) instead of actual precipitation recorded in the study area or nearby. Is this kind of information reliable? Why not a small portable weather station to record instant in-situ parameters? Or at least observed information from nearby weather stations?

The observer used the weather application during the observations itself for ease of categorising weather. A common way that weather is judged across the population.  The observer had to travel to the study site on a pedal bicycle into the middle of Richmond Park. Carrying further equipment was not really an option.

Results

Expand the information in tables and figure captions to better understand the source of data. For example, I suppose results presented in Table 2 and Figures 4, 5 are from the GEEs (and not from univariate models). I suggest to summarize the outcome tables from GEE models in supplementary material if not presented in the main text (maybe too much numbers), with coefficient estimates, errors, statistics, p-values, etc.

I do not understand the comment here. We have provided the output for the modelling and we have included supplementary information on what raw data were calculated for all behaviours in the supplementary information. We have been clear and open in our model outputs.

 L286-295 – This result shows an interaction between social period and sex predictors on natural foraging. Authors could consider first order interactions in the models, though that could increase complexity and convergence problems in some cases.

Thank you for the comment. We have already edited the manuscript, based on comments from other reviewers, the potential effects of sex and development of eggs in female birds during the pre-laying period as a reason for the change in behaviour.   

L306-308 – Information related to predictor description (e.g. weather categories) must be included in Methods’ section and not in Results. Also, provide more details on how each category is defined (e.g. cloud coverage thresholds).

Thank you for the comment. We have already included this in the method, and we have provided a reference. This was a description from the weather service used and we have used their standard descriptions of weather.  

Discussion

The discussion is complete, neat, and well supported by previous knowledge and referenced literature. Concerns could arise from discussion of problematic predictors, such as pond size.

Thank you for the comment on the discussion.

We wish to keep in pond size because they were different in size and degree of cover and how many people who visited the park visited each pond. We have been clear in description of the habitat and we have been clear in our discussion that this is a study on one population and that others can extend it if they wish, based on the information that we have provided.  

L346-349 – Higher foraging activity could be related also to higher energy demand during the breeding seasonal, not only to higher food availability. In fact, evolutionary pressures would lead to synchronized highly demanding breeding activities and resource pulses in seasonal environments. This could explain also why females showed higher natural foraging rates during laying and pre-laying periods, much stronger than in males.

Thank you for the comments. We have already included this information in the discussion, on differences in energetics due to the laying period and development of eggs.

L374-386 – Although most of the interpretations could be reasonable according to the species ecology, inference on pond size effects based on only to small non-independent sites with different habitat structure (e.g. vegetation coverage, islands) is not possible as already commented in Methods. At most, those can be collateral non-tested conjectures.

Thank you for the comments. Given the differences in the size of the ponds and their location within the Park, we feel it reasonable that this could affect habitat choice. We have included a reference to show local (small scale) habitat choices are possible, therefore we feel that these two ponds are worthy of study.

Reviewer 4 Report

In this paper, the two researchers aimed to use observations on wild birds to inform husbandry options for captive species. This is a valuable aim and the paper presents useful information on the behaviour of one species that is often reared in captive conditions, the mandarin duck. The paper is well-written and carefully analyzed. My one concern is that when I saw that the behaviour of captive and wild animals was compared, I imagined that researchers went to study the species in the wild. But it turns out the wild is in central London in managed ponds with public access. Perhaps an alternative to wild like semi-natural would convey a more accurate picture. The following comments are minor and can be easily addressed.

Line 61: Perhaps charismatic might be better term than enigmatic.

Data collection: There was no mention of rings on the birds so I assume that it was not possible to identify individual birds. Therefore, it is likely that the same birds were monitored repeatedly during the study period. I see that steps were taken to reduce selection bias. Perhaps be a little clearer about pseudo-replication and its potential consequences.

Table 1: As per Beauchamp’s (2015) book on vigilance, the definition of vigilance in the first part of table 1 seems concerned with reactive vigilance. Later the event ‘alert’ is described, which appears concerned with pre-emptive vigilance. You may want to rethink these labels. Not many readers will see a difference between vigilance and alertness.

Line 171: How did you define a pair? Just to be clear, data were collected for the two members of a single pair for 20 minutes, correct? Presumably the information was used to build the hourly time budgets that we see later in the figures. Was there only one bout of observations per h to fit in these hourly slots?

Line 199: Did you record group size, the number of conspecifics within a given distance of the focal birds? This is an important variable as it can influence feeding and vigilance. Perhaps density might be a better name since groups are probably unlikely at this time of year.

Line 206: At the moment, it was not clear to me where the data from the wild and from captive populations came from. The Richmond data must be considered semi-natural because of possible feeding and presumably lack of predation. A brief description of the two sources of data from the onset might clarify the matter.

Table 2: I The labels for weather suggest that at least some of these variables were not quantitative (e.g. light cloud). These should be explained. I also see a label called Mandarin No. This has not been defined earlier, I believe.

Line 337: It would be better I think to mention the key environmental variables influencing time budget rather than listing all variables measured here. For instance, there was no effect of time of day.

Discussion: Are these ducks under any significant threats from starvation or from predation? In Taiwan, the sex ratio is biased towards males and predation rate especially on females appears quite large (Sun, Y.-H., C. L. Bridgman, H.-L. Wu, C.-F. Lee, M. Liu, P.-J. Chiang, and C.-C. Chen (2011). Sex ratio and survival of Mandarin Ducks in the Tachia River of Central Taiwan. Waterbirds 34:509-513). These considerations are important when interpreting their behaviour.

Line 346: The difference between these two estimates is rather small (about 5%) and it comes from different studies. Females at least might need to accumulate reserves for egg laying. Could this explain seasonal differences? Also, there are probably differences in food types (vegetation v. animals) that could be involved.

Line 452: I do not recall a significant effect of mandarin numbers (density perhaps?) on vigilance. An increase in vigilance with density need not indicate that birds are doing the same as conspecifics. Typically, researchers assume that neighbours can pose threats to which individuals respond by increasing vigilance (e.g. Beauchamp, G. (2016). Function and structure of vigilance in a gregarious species exposed to threats from predators and conspecifics. Animal Behaviour 116:195-201). The same goes for an increase in social interactions, which may not be positive at all. An increase in feeding might indicate more competition. I agree that social facilitation can be important in some contexts, but given that we have alternatives based on ecology, is there a need to invoke this here?

Line 497: I would not necessarily conclude that what we saw in mandarins is aimed at maximizing survival and reproduction. These birds were observed in semi-natural conditions after all.

Author Response

In this paper, the two researchers aimed to use observations on wild birds to inform husbandry options for captive species. This is a valuable aim and the paper presents useful information on the behaviour of one species that is often reared in captive conditions, the mandarin duck. The paper is well-written and carefully analyzed. My one concern is that when I saw that the behaviour of captive and wild animals was compared, I imagined that researchers went to study the species in the wild. But it turns out the wild is in central London in managed ponds with public access. Perhaps an alternative to wild like semi-natural would convey a more accurate picture. The following comments are minor and can be easily addressed.

Thank you for the comment and the kind words on the paper. These mandarins are free-living. We have edited the title accordingly. We have explained that free-living refers to wild birds in the text.

Please note that these birds are not managed and come and go as they please. They do not spend all of their time within the wooded wetland and they fly in and out as they wish, in the same way as the mallards and other wildfowl do. They are wild individuals within a naturalised population.

Line 61: Perhaps charismatic might be better term than enigmatic.

Edited

Data collection: There was no mention of rings on the birds so I assume that it was not possible to identify individual birds. Therefore, it is likely that the same birds were monitored repeatedly during the study period. I see that steps were taken to reduce selection bias. Perhaps be a little clearer about pseudo-replication and its potential consequences.

We have included a comment on pseudoreplication in the methods and into the discussion too, as a potential research extension (ringing the birds to allow for individual identification).

Table 1: As per Beauchamp’s (2015) book on vigilance, the definition of vigilance in the first part of table 1 seems concerned with reactive vigilance. Later the event ‘alert’ is described, which appears concerned with pre-emptive vigilance. You may want to rethink these labels. Not many readers will see a difference between vigilance and alertness.

Thank you for the comment. We have clarified (and added detail to) alert as an event behaviour in our ethogram to ensure others can follow.

Line 171: How did you define a pair? Just to be clear, data were collected for the two members of a single pair for 20 minutes, correct? Presumably the information was used to build the hourly time budgets that we see later in the figures. Was there only one bout of observations per h to fit in these hourly slots?

The identification of mandarin pairs from their behaviour has been included in the methods.

There were multiple 20-minute observations per morning, midday and afternoon time period. This has been clarified in the text although it was already included in the behavioural observations section.

Line 199: Did you record group size, the number of conspecifics within a given distance of the focal birds? This is an important variable as it can influence feeding and vigilance. Perhaps density might be a better name since groups are probably unlikely at this time of year.

This is explained in section 2.2. The number of mandarins and other birds was recorded as the start of each 20 minute observation as an overall total. Then the observer completed the 20 minutes, then repeated the count of birds, and then did another 20 minutes on new ducks.  

Line 206: At the moment, it was not clear to me where the data from the wild and from captive populations came from. The Richmond data must be considered semi-natural because of possible feeding and presumably lack of predation. A brief description of the two sources of data from the onset might clarify the matter.

We have included information on the size and scale of Richmond Park and the natural environment. More references to this have also been included.

The birds in Richmond Park are sometimes fed by the public in the same manner as wild mallards or Canada Geese are but feeding of the wildlife is actively discouraged by Royal Parks. These mandarins are wild. They are not managed, they are subject to predation by foxes, birds of prey and the usual suite of British predators. Richmond Park is a huge space and the birds move freely within it. We have stated the description of the Park and the specific woodland we used because of the scale of the area.

We have explained, multiple times in the text that the captive data are published in Bruggers, R. L., & Jackson, W. B. (1977). Time budgets of Mandarin Ducks under semi-natural conditions. Wildfowl28(28), 7. This paper is on a captive population of pinioned birds.

Table 2: I The labels for weather suggest that at least some of these variables were not quantitative (e.g. light cloud). These should be explained. I also see a label called Mandarin No. This has not been defined earlier, I believe.

Weather description was taken from BBC Weather. This is explained already in section 2.2.

Again, duck number is explained in section 2.2. this was recorded at the start of each 20 minute observation.

Line 337: It would be better I think to mention the key environmental variables influencing time budget rather than listing all variables measured here. For instance, there was no effect of time of day.

My apologies, but I do not understand the comment. We have explained each potential predictor in turn and this is mirrored in the discussion. We have clearly stated there was no effect of time of day.

Discussion: Are these ducks under any significant threats from starvation or from predation? In Taiwan, the sex ratio is biased towards males and predation rate especially on females appears quite large (Sun, Y.-H., C. L. Bridgman, H.-L. Wu, C.-F. Lee, M. Liu, P.-J. Chiang, and C.-C. Chen (2011). Sex ratio and survival of Mandarin Ducks in the Tachia River of Central Taiwan. Waterbirds 34:509-513). These considerations are important when interpreting their behaviour.

Thank you for the comment. These are wild birds that are at risk of predations. This has been included in the text.

Line 346: The difference between these two estimates is rather small (about 5%) and it comes from different studies. Females at least might need to accumulate reserves for egg laying. Could this explain seasonal differences? Also, there are probably differences in food types (vegetation v. animals) that could be involved.

Thank you for the comment. We have changed high to higher. We already state the seasonality in food availability but we have included animal-based food items specifically and the energetic needs of the female bird.

Line 452: I do not recall a significant effect of mandarin numbers (density perhaps?) on vigilance. An increase in vigilance with density need not indicate that birds are doing the same as conspecifics. Typically, researchers assume that neighbours can pose threats to which individuals respond by increasing vigilance (e.g. Beauchamp, G. (2016). Function and structure of vigilance in a gregarious species exposed to threats from predators and conspecifics. Animal Behaviour 116:195-201). The same goes for an increase in social interactions, which may not be positive at all. An increase in feeding might indicate more competition. I agree that social facilitation can be important in some contexts, but given that we have alternatives based on ecology, is there a need to invoke this here?

Thank you for the comment. We have added in the significant outputs from the model for social interaction and vigilance that are mentioned in the discussion. Apologies for the oversight.

Line 497: I would not necessarily conclude that what we saw in mandarins is aimed at maximizing survival and reproduction. These birds were observed in semi-natural conditions after all.

Thank you for the comment and apologies for the repetition but these are wild birds. These birds are not semi-natural. They are free-living, the same as the mallards, Canada geese and moorhens that they are sharing the habitat with.

Round 2

Reviewer 1 Report

Dear authors,

Thank you for kindly addressing all my comments. I consider your work suitable for publication.

Kind regards

Author Response

Thank you for the helpful and developmental comments on the manuscript. 

Reviewer 2 Report

Authors addressed all of my previous comments and provided me with a detailed rebuttal letter. Therefore, in my opinion, the manuscript can be accepted for publication in its current form.

Author Response

Thank you for your helpful and developmental comments on the manuscript. 

Reviewer 3 Report

Ms-ID: animals-1860073

Second Review

Environmental and social influences on the behaviour of wild mandarin ducks in Richmond Park

Camille Munday, Paul Rose

Authors have corrected mistakes and incorporated extra information to clarify some concerns pointed out in the last review. There are still some important points to be addressed related to methods.

Pond size - With the current experimental design, it is not possible to strictly test the effect of pond size. There are no replicates for each level of size (small/large), just to non-independent ponds (250 m away). Therefore, use “pond id” as predictor name instead of “pond size” throughout the manuscript (including tables and figures). Possible pond size effects can still be discussed as currently done by comparing both individual ponds, though those will not be generalizable interpretations but plausible conjectures that cannot be disentangled from other correlated features as vegetation coverage (Peg’s pond is larger and with higher vegetation coverage). The extended rationale on the study limitations and the suggestion of further works with a better sampling design is fine.

Check homocedasticity (L249-250) – When two independent samples are compared with a t-Student test, the homogeneity of variance assumption must be tested (e.g. Leven’s test) in addition to normality. If variances are not equally distributed then a non-parametric test can be used instead (as done with Activity and Inactivity behaviors).

GEEs and pseudorreplication - The application of GEEs on long dataset with repeated measures is perfectly fine. But if you do not have a factor identifying subjects repeatedly sampled (e.g. bird, pond, etc.), then you are basically performing an equivalent simple GLM without controlling for potential pseudoreplication. When running GEE in SPSS, one of the first steps in the Repeated Windows is to select Subject Variable, which is the factor identifying the sample unit measured repeatedly ( random factors in GLMM), and to select Within-Subject Variables, that is, those variables accounting for different sampling events. In your study, birds were not individually banded so bird ID cannot be used as the Subject Variable. That was why I asked if you were using pond Identity as the Subject Variable, since multiple repeated observations were done in each pond. Anyway, authors state that the study did not control for individual contribution and thus they cannot discard biased estimates due to overrepresented individuals.  

GEEs and correlation structure - I still recommend to use another working correlation structure more similar to the actual design instead of the exchangeable structure used, which assumes the same fixed correlation value among observations. For example, the first order autoregressive AR(1) that set a different (usually stronger) correlation value between consecutive observations. It is true that for balanced datasets with few missing values, the parameter estimates do not much change, and only the standard error (and thus p-values) can show variation.

Other minor corrections:

L110 – Replace “told” with “total”.

L116 – I think the specific names of both ponds should be introduced here.

L182-183 (Table 1) – “Contraspecific”? Do you mean interspecific interaction?

L223 – “and” instead of “agnd”.

L226 – Remove “Northern”, only “Mallard”.

Figfure 3 - If Mandarins did not devote any time to terrestrial locomotion and reproduction -(i.e. 0% ± 0; L181-182), why did those behaviors show a small (1%) but different from zero percentage of time in Figure 3? Correct the Figure or values in Results.

L502 – Do you mean “conspecific interactions”? You are discussing the effect of Mandarin numbers on Mandarin behaviors, not the effect of other bird species.

Author Response

Authors have corrected mistakes and incorporated extra information to clarify some concerns pointed out in the last review. There are still some important points to be addressed related to methods.

Pond size - With the current experimental design, it is not possible to strictly test the effect of pond size. There are no replicates for each level of size (small/large), just to non-independent ponds (250 m away). Therefore, use “pond id” as predictor name instead of “pond size” throughout the manuscript (including tables and figures). Possible pond size effects can still be discussed as currently done by comparing both individual ponds, though those will not be generalizable interpretations but plausible conjectures that cannot be disentangled from other correlated features as vegetation coverage (Peg’s pond is larger and with higher vegetation coverage). The extended rationale on the study limitations and the suggestion of further works with a better sampling design is fine.

Thank you for the comment. We have edited to Pond ID.

Check homocedasticity (L249-250) – When two independent samples are compared with a t-Student test, the homogeneity of variance assumption must be tested (e.g. Leven’s test) in addition to normality. If variances are not equally distributed then a non-parametric test can be used instead (as done with Activity and Inactivity behaviors).

Thank you for the comment. We have checked with our stats advisors and they provided this answer.

You created QQ plot to see whether these data were normally distributed. You only need to perform a Levene's Test if you want to compare 2(+) groups on a quantitative variable, to see if they have equal scores. As you are comparing separate groups, you can run an independent samples t-test  (which is what we have done). The tests are listed below.

 Test 1. Wild Pre-Incubation Loaf Versus Captive Pre-Incubation Loaf

Test 2. Wild Laying Loaf Versus Captive Laying Loaf

Test 3. Wild Post-Incubation Loaf Versus Captive Post-Incubation Loaf

 Test 4. Wild Pre-Incubation Forage Versus Captive Pre-Incubation Forage 

Test 5. Wild Laying Forage Versus Captive Laying Forage 

Test 6. Wild Post-Incubation Forage Versus Captive Post-Incubation Forage 

Consequently we have not changed this testing.

GEEs and pseudorreplication - The application of GEEs on long dataset with repeated measures is perfectly fine. But if you do not have a factor identifying subjects repeatedly sampled (e.g. bird, pond, etc.), then you are basically performing an equivalent simple GLM without controlling for potential pseudoreplication. When running GEE in SPSS, one of the first steps in the Repeated Windows is to select Subject Variable, which is the factor identifying the sample unit measured repeatedly (≈ random factors in GLMM), and to select Within-Subject Variables, that is, those variables accounting for different sampling events. In your study, birds were not individually banded so bird ID cannot be used as the Subject Variable. That was why I asked if you were using pond Identity as the Subject Variable, since multiple repeated observations were done in each pond. Anyway, authors state that the study did not control for individual contribution and thus they cannot discard biased estimates due to overrepresented individuals.  

We have clarified further in the text that date of the observation was the random factor included in the GEE.   

GEEs and correlation structure - I still recommend to use another working correlation structure more similar to the actual design instead of the exchangeable structure used, which assumes the same fixed correlation value among observations. For example, the first order autoregressive AR(1) that set a different (usually stronger) correlation value between consecutive observations. It is true that for balanced datasets with few missing values, the parameter estimates do not much change, and only the standard error (and thus p-values) can show variation.

Thank you for the comment. We feel that this would need a substantial re-write of the paper and a change of statistical approach compared to the one that we have used, was provided by experts in the field and has already been assessed and marked based on the student's piece of original research work that was submitted. If the reviewer feels this strongly about the analysis then the paper will need to be substantially revised and the editor provides a more reasonable amount of time for this to occur. We however, have acknowledged all methodological limitations, sought appropriate help with analyses and are confident in our findings. 

Other minor corrections:

L110 – Replace “told” with “total”.

Edited

L116 – I think the specific names of both ponds should be introduced here.

Edited

L182-183 (Table 1) – “Contraspecific”? Do you mean interspecific interaction?

Edited

L223 – “and” instead of “agnd”.

Edited

L226 – Remove “Northern”, only “Mallard”.

Edited

Figfure 3 - If Mandarins did not devote any time to terrestrial locomotion and reproduction -(i.e. 0% ± 0; L181-182), why did those behaviors show a small (≈1%) but different from zero percentage of time in Figure 3? Correct the Figure or values in Results.

We have clarified these results and edited the text to ensure the reader can follow what data are presented. We have provided further discussion for these low values and a further research extension point. 

L502 – Do you mean “conspecific interactions”? You are discussing the effect of Mandarin numbers on Mandarin behaviors, not the effect of other bird species.

Edited 

Reviewer 4 Report

Thank you for taking my comments into account. I have no further comments.

Author Response

Thank you for your consideration of the manuscript.

Round 3

Reviewer 3 Report

Ms-ID: animals-1860073

Third Review

Environmental and social influences on the behaviour of wild mandarin ducks in Richmond Park

Camille Munday, Paul Rose

Homocedasticity assumption – I now understand that, for foraging and loafting, you compared the mean value of sampled wild birds against a single mean score for captive birds from literature (dataset not available) using a one-sample t-test (not a two independent samples). Then it is ok to check only normality and not homocedasticity.

GEEs and pseudorreplication – GEEs do not account on random effects (subject effects), but on averaged estimates across population (sample). Random effects/factors are thus term used in (G)LMM, not in GEE. I supposed you included Time as a Within-subject variable in the SPSS model, which is the variable accounting for the repeated sampling events. Change the term “random factor” (L261-262).

I guess you did not set the Subject Variable (on which repeated measures are made) because you do not have bird IDs. That is why you do not control for pseudorreplication as discussed and explained in the manuscript.

GEEs and correlation structure – Your data set consists on sequential observations. It is therefore reasonable to think that consecutive observations are strongly correlated than distant ones. Using a fixed correlation value (exchangeable structure) means that all all observations are equally correlated, which does not make much sense. It is more realistic to use, for example, an autoregressive AR(1) structure that set a higher correlation value between consecutive observations. Changing the correlation structure is easy and immediate if you have the same data set arrangement used for the original analysis. You need only to change one option in the SPSS model window. Parameters estimates will be very similar if the data set is balanced and with few missing values, though SE and associated p-values may change. If effects are strong enough, the outcomes and their interpretation will be the same. I recommend trying this correlation structure, otherwise authors should justify the use of the exchangeable structure.

Minor corrections:

L283-286 – I suggest to merge and shorten both sentences as (for example): “… which occurred the least (Fig. 3), while flying was observed 0% ±0 of the time with an average number of flight events of ## per 20 minutes (range: 0-6).”

Table 2 – Replace “Pond size” with “Pond ID”.

L555 – I am not sure the following sentence is grammatically correct: “to fully determine how often, and for how long, ducks spend flying.”. Should it be “ducks fly”?

Author Response

Homocedasticity assumption – I now understand that, for foraging and loafting, you compared the mean value of sampled wild birds against a single mean score for captive birds from literature (dataset not available) using a one-sample t-test (not a two independent samples). Then it is ok to check only normality and not homocedasticity.

No edit required

GEEs and pseudorreplication – GEEs do not account on random effects (subject effects), but on averaged estimates across population (sample). Random effects/factors are thus term used in (G)LMM, not in GEE. I supposed you included Time as a Within-subject variable in the SPSS model, which is the variable accounting for the repeated sampling events. Change the term “random factor” (L261-262).

We are unsure of what you are asking for here as the edit is unclear. We have removed random factor and changed to "... was included to account for repeated sampling events". 

I guess you did not set the Subject Variable (on which repeated measures are made) because you do not have bird IDs. That is why you do not control for pseudorreplication as discussed and explained in the manuscript.

No edit required.

GEEs and correlation structure – Your data set consists on sequential observations. It is therefore reasonable to think that consecutive observations are strongly correlated than distant ones. Using a fixed correlation value (exchangeable structure) means that all all observations are equally correlated, which does not make much sense. It is more realistic to use, for example, an autoregressive AR(1) structure that set a higher correlation value between consecutive observations. Changing the correlation structure is easy and immediate if you have the same data set arrangement used for the original analysis. You need only to change one option in the SPSS model window. Parameters estimates will be very similar if the data set is balanced and with few missing values, though SE and associated p-values may change. If effects are strong enough, the outcomes and their interpretation will be the same. I recommend trying this correlation structure, otherwise authors should justify the use of the exchangeable structure.

Thank you for the comment and we appreciate your clear knowledge of statistical analysis. We are keeping the model as it. We attempted to choose different birds. The observer randomly selected individuals so as to not provide consecutive observations of the same animals. And we are trying to see difference between behaviour with habitat (for example) so we are assuming no difference between animal responses (hence equal correlation between all potential dependent variables). We explained how birds were selected, over what time period and at what pond, and how the observer randomly selected birds to minimise repetition as best possible. We are confident as best we can that we have minimised consecutive observations of the same animal. All of this information was provided to the University's statistics department and all of their advice was followed on how to build and apply the final GEE. 

Minor corrections:

L283-286 – I suggest to merge and shorten both sentences as (for example): “… which occurred the least (Fig. 3), while flying was observed 0% ±0 of the time with an average number of flight events of ## per 20 minutes (range: 0-6).”

We will keep this sentence as is.

Table 2 – Replace “Pond size” with “Pond ID”.

Edited

L555 – I am not sure the following sentence is grammatically correct: “to fully determine how often, and for how long, ducks spend flying.”. Should it be “ducks fly”?

Edited